# Psychological Impact of NBS for CF

**DOI:** 10.3390/ijns6020027

**Published:** 2020-03-30

**Authors:** Jane Chudleigh, Holly Chinnery

**Affiliations:** 1School of Health Sciences, City, University of London, London EC1V 0HB, UK; 2Faculty of Sports, Health and Applied Science, St Mary’s University, London TW1 4SX, UK; holly.chinnery@stmarys.ac.uk

**Keywords:** newborn bloodspot screening, cystic fibrosis, psychological impact

## Abstract

Newborn screening for cystic fibrosis has resulted in diagnosis often before symptoms are recognised, leading to benefits including reduced disease severity, decreased burden of care, and lower costs. The psychological impact of this often unsought diagnosis on the parents of seemingly well children is less well understood. The time during which the screening result is communicated to families but before the confirmatory test results are available is recognised as a period of uncertainty and it is this uncertainty that can impact most on parents. Evidence suggests this may be mitigated against by ensuring the time between communication and confirmatory testing is minimized and health professionals involved in communicating positive newborn screening results and diagnostic results for cystic fibrosis to families are knowledgeable and able to provide appropriate reassurance. This is particularly important in the case of false positive results or when the child is given a Cystic Fibrosis Screen Positive, Inconclusive Diagnosis designation. However, to date, there are no formal mechanisms in place to support health professionals undertaking this challenging role, which would enable them to meet the expectations set out in specific guidance.

## 1. Introduction

The increased use globally of newborn bloodspot screening (NBS) for cystic fibrosis (CF) has resulted in diagnosis often before symptoms are recognised. Benefits include reduced disease severity, decreased burden of care, and lower costs [1,2,3,4,5]. Without NBS, the diagnosis of CF relies on the recognition of particular clinical signs and symptoms, which often results in delayed diagnosis. This can lead to an arduous journey for parents characterised by uncertainty and anxiety as they seek answers and are referred to a number of different clinical specialities before the correct, definitive diagnosis is made [6].

NBS for CF may pose different challenges when compared to other conditions included in NBS programmes, such as sickle cell disease (SCD) which commonly includes antenatal screening, meaning that parents are aware of their own carrier status and the theoretical risk to their unborn child [7]. For CF, parents are often unaware and have not sought information regarding their own carrier status [8]. However, other challenges, such as the method and content of communication, may be similar between conditions.

Despite the undisputed clinical and fiscal benefits of CF NBS, several challenges have been noted, one of which being the potential psychological impact on the child’s family. One small study consisting of qualitative interviews with the parents of children diagnosed with CF either via NBS, prenatally, or after the development of symptoms suggested that for those diagnosed via NBS, the early, unsought diagnosis had the potential to deeply affect parents in many ways. These included questioning their competence to care for their baby and their sense of who the baby is. In addition, early diagnosis led to the disease taking centre stage during the child’s early weeks and months and caused health professionals to loom very large in the family’s life at this formative time [9]. Another study in the US, which explored the parental experiences associated with receiving a positive NBS result for one of the metabolic conditions, supported these findings and suggested that the methods used to communicate the NBS result and the condition specific knowledge of the individual imparting the result influenced parental dissatisfaction, anxiety, and distress [10]. The similarities between the findings of these studies perhaps reflect the fact that CF and the metabolic conditions have a genetic origin and therefore staff knowledge and understanding about the cause and immediate and longer term implications of these are vital.

Like all conditions, it is important that careful consideration is given to how positive CF NBS results are communicated to parents as this is often unexpected, represents a life limiting diagnosis for the child and often a life changing event for the parents. As for many conditions, it may not be possible to remove parental anguish completely from what is an upsetting time. However, it is important for health professionals to communicate positive CF NBS results in a manner that minimises potential distress and does not detrimentally affect parents’ relationships with their child and other family members. This chapter will focus on the psychological impact of CF, with implications not always limited to CF, and will explore the current guidance regarding communication strategies, the impact of poor communication practices, and information giving in times of uncertainty, and make recommendations for future practice.

## 2. Guidance Regarding Communication of Positive NBS Results for CF

Internationally, detailed guidelines exist for the processing of positive CF NBS results [11,12]. However, these primarily focus on laboratory processes and subsequent clinical management; less attention is given to how positive CF NBS results should be communicated to families to minimise potential distress. The European best practice guidelines for CF neonatal screening [3] recognise the time period during which the screening result is communicated to families but before the confirmatory test results are available, as a “period of maximal uncertainty.” It is therefore suggested that during this time, information should be provided to families in a format that will be easy for them to digest, using language the family can understand. In addition, information should be structured, clarified, and summarised appropriately and parental understanding should be checked and questions encouraged. These guidelines also suggest the health professional communicating the NBS result should explore the families’ beliefs, concerns, and expectations in order to tailor information and the conversation style for the needs of the parent(s). Moreover, in anticipation of further communication needs, the health professional should encourage parental participation in decisions and enlist resources and appropriate support. Finally, health professionals informing parents of their child’s positive CF NBS result should be knowledgeable about CF, NBS principles, basic CF genetics, and the psychosocial pitfalls that some parents may experience [3]. 

More specific guidance from the United Kingdom states that families should be informed by an initial structured telephone call undertaken by a well-informed health professional with appropriate experience and support to give bad news [13]. European guidelines recognise the importance of CF team members possessing compassionate communication and effective information giving skills and the ability to recognise and respond to emotional distress. In addition, these guidelines suggest that some CF team members may require training in more specific skills, such as breaking bad news, recognising significant psychopathology, and appropriate referral should such instances occur. The advantages of the inclusion of specialist mental health professionals in CF teams, such as clinical psychologists or psychiatrists, is also recognised [12]. 

## 3. Impact of Communicating Positive NBS Results to Families

In 2015, a systematic review summarised if, and how, information provision has been included in economic evaluations of NBS programmes [14]. This review highlighted that only three studies included an estimate of the cost of information provision in their analysis and none of the studies captured the impact of information provision after screening [14]. One study in the systematic review [14] referred to costs related to the impact of poor information provision specifically related to false positive results rather than poor information provision at the time of communicating the initial NBS+ result per se [15]. This review concluded that evidence existed to support the notion that poor information provision in relation to NBS does impact on parents but there have been few attempts to quantify the impact of information provision in economic evaluations of NBS to date. Importantly, this review confirmed that there are no current data on the long-term impact of poor information provision and subsequent use of healthcare resources and impact on parents’ health and well-being. This is interesting since the provision of adequate information and therefore good parental knowledge about their child’s false positive CF NBS result, meant that the number of visits to the child’s General Practitioner did not differ significantly between the false positive and the negative NBS groups [16]. Therefore, there is clearly a need to explore the role of information provision on the subsequent healthcare resource use. 

Studies that have focused on CF NBS have identified adverse outcomes associated with the approaches and methods used to communicate CF NBS results to families. Interviews with the mothers (*n* = 106) and fathers (*n* = 97) of children with a confirmed diagnosis of congenital hypothyroidism (CHT) (*n* = 37), CF-carrier (*n* = 43), or CF (*n* = 26) in the United States (US) found that the majority of parents across all groups reported strong initial emotional reactions such as shock, panic, and anxiety about what results meant. The responses are likely related to the fact that currently, antenatal screening for CHT and CF is not routinely undertaken, therefore parents are unaware of the potential risk of their unborn baby being affected by these conditions. Responses related to positive CF NBS results included fears of the child dying, parental somatic symptoms, such as nausea and suffocation, difficulty bonding with their infant, marital discord, and changed reproductive plans [17]. These differences may reflect the fact that while CHT is treatable with a very good prognosis when diagnosed and treated in infancy, CF continues to be a life-limiting condition with no cure. In addition, in the majority of cases, CHT does not have a genetic origin and therefore does not have the same reproductive implications as CF. A similar study conducted in the US explored factors affecting parent–child relationships one year after positive NBS with 131 mothers and 118 fathers of 131 infants who had a positive NBS result for CF (*n* = 23), CF carrier (*n* = 38), CHT (*n* = 35), or normal NBS (*n* = 35). The parents of children with CF reported higher perceptions of child vulnerability and fathers of children who were CF carriers, viewed their children as more attached. The findings also indicated that infant feeding problems, particularly in the presence of a serious health condition like CF, could represent an important sign of more deeply rooted concerns regarding the parent–child relationship [18]. 

A study in the UK to explore parents’ experience of receiving a positive NBS result used semi-structured interviews with 12 mothers (five with a child with CF and seven with a child with SCD and 10 fathers (five each with a child with CF or SCD) of children diagnosed via NBS [19]. The mothers of infants with a positive NBS for CF found being alone when they received their child’s positive CF NBS result upsetting and fathers expressed distress at not being able to support their partner during this time. This also reportedly had the potential to impact on parental relationships as the mother then became responsible for informing the baby’s father of the positive CF NBS result. These findings were not reported by the parents of babies who had received a positive NBS for SCD who described being aware of their ‘risk’ due to the results of antenatal screening [19]. Therefore they were less shocked by the result but were more concerned with the stigma associated with a diagnosis of SCD, which has been commonly cited in the literature [20]. Conversely, the parents of babies with a positive NBS results for CF did not report feeling stigma associated with the condition. CF NBS results also impacted on parental relationships in other ways, including parents questioning their choice of partner and feelings of confusion and guilt at having passed the defective gene on to their child. This was similar to the responses of parents whose baby had received a positive NBS for SCD and perhaps reflects the genetic implications of both conditions.

Receiving the CF NBS result from a health professional perceived to be less informed and therefore unable to answer parental questions about CF was also undesirable and had the potential to impact on future relationships with that health professional [19]. It should be noted that the sample size in this study was small but reflects the findings of other studies. A prospective questionnaire survey of 138 parents who had received a positive CF NBS in Switzerland indicated that most parents (*n* = 122; 88%) were satisfied with screening, four (3%) were not, and 12 (9%) were unsure. The parents received their child’s positive CF NBS result over the telephone from a CF physician and were invited for diagnostic testing during the same conversation; 100 (74%) of the parents found the information provided satisfactory. This supports the importance of the person reporting the NBS result having condition specific knowledgeable. The remaining parents who were unsatisfied stated the caller had not explained the test result and the disease or had provided superficial information and instead focused on arranging the appointment [21].

A study in Germany evaluating CF NBS since its introduction in 2016 found that of the 105 (54.7%) families involved in the study (out of 192 who had gone through diagnostic testing after a positive CF NBS result), only 30 parents obtained information about the newborn screening by a physician despite this being mandatory in Germany. Despite this, parents being informed about the positive CF NBS result by a CF specialist were more satisfied with the given information than those informed by the maternity ward. Furthermore, waiting for more than 3 days between the information about the CF NBS result and the diagnostic testing was too long for 77.7% of the families [22]. These findings and those of the study in Switzerland [21] highlight the importance of ensuring that diagnostic testing is undertaken in a timely manner to reduce the parental anxiety and uncertainty associated with the positive NBS result.

Evidence also exists regarding the impact of communicating practices specifically for NBS carrier results. Semi-structured face-to-face interviews conducted with 49 mothers, 16 fathers and 2 grandparents of 51 infants identified CF NBS as carriers of CF (*n* = 27) and SCD (*n* = 24), in England demonstrated untoward anxiety or distress among parents was influenced by how results were conveyed rather than the carrier result per se [23].

In summary, communication of positive CF NBS can influence outcomes in the short term [17,19,21,23] but may also have a longer term impact on children and families [18]. Evidence suggests that the distress caused can manifest in several ways, including arguments between couples, including the apportioning of blame [19,23], the alteration of life plans, an inability to conduct the tasks of daily living, such as going to work or socializing [23], long-term alterations in parent–child relationships [18], and mistrust and lack of confidence affecting ongoing relationships with staff [19].

## 4. Dealing with Uncertainty: False Positives and CF Screen Positive, Inconclusive Diagnosis (CFSPID)

It has already been highlighted that the time period during which the NBS result is communicated to families but before the confirmatory test results are available, is a “period of maximal uncertainty” [3]. The impact of uncertainty associated with NBS results has been considered extensively in the literature and similar issues have been identified for many of the conditions included in NBS programmes. This is an important consideration that is by no means exclusive to the CF community but may be more prevalent in conditions with a genetic origin such as CF, SCD, and the metabolic conditions due to the longer term implications such as the effect on future reproductive decisions [24,25,26].

False positive NBS results have the potential to lead to ongoing uncertainty for parents. However, engaging the parents of children who have received a false positive results for any of the conditions included in NBS programmes in research, is notoriously difficult. Studies that have managed to capture this study population have produced conflicting information regarding the potential for false positive results to have a detrimental effect on the family system.

In France, a prospective study conducted with 86 families at 3, 12, and 24 months after receiving a false positive CF NBS result using the Perceived Stress Scale, and the Vulnerable Child Scale found that although 96.5% of parents said they had been anxious at the time of the sweat test, 86% felt entirely reassured 3 months afterwards. The mean perceived stress scale scores did not differ from the French population and the mean vulnerable child scale indicated a low parental perception of child vulnerability. These scores were not found to differ at 12 and 24 months after receiving the false positive CF NBS result. Indeed, 86% to 100% of families no longer worried about CF and all parents stated that they would have the test performed again for another child [27].

Similarly, in the Netherlands, 62 parents (59%) who had received a false positive CF NBS result, and 146 parents (46%) who had received a negative CF NBS result, returned questionnaires to assess long-term effects of false positive results on parental anxiety and stress. In addition, 24 mothers and three fathers participated in 25 semi-structured interviews. Parents showed strong negative feelings after being informed about the positive CF NBS screening result and satisfaction with time of referral was negatively associated with the number of days between being informed and the appointment at the hospital (*r* = 0.402; *p* = 0.001), indicating the importance of timely confirmatory testing. After confirmation that their child was healthy and not affected by CF, most parents felt reassured. Indeed, parental concern about their baby’s health or the number of visits to their General Practitioner did not differ significantly between the false positive and the negative NBS groups. After six months, no difference in anxiety levels between both groups of parents was found. Only 6% (4/62) of parents who received a false positive CF NBS result said they would not participate in NBS in the future, while 16% (10/62) were not sure [16].

Likewise, a study in Canada that included 134 mothers who had received a false positive CF NBS for their child and 411 controls who completed questionnaires when their infant was 2 and 12 months old and 54 mothers who had received a false positive CF NBS for their child who were interviewed found that mean anxiety, distress, and vulnerability scores did not differ between the two groups. Of those who received a false positive CF NBS result, 61% were informed by their primary care physician and 39% by a genetic counsellor. The majority (87%) of mothers stated the time between being notified of the positive screen and learning the final results was the “scariest time of their lives”, stating that having been home from hospital with an apparently healthy infant, it was alarming to learn that their child might have a chronic illness. Mothers placed tremendous value on the fact that time to confirmatory testing was quick (generally ≤48 h) and valued the excellent coordination of care, particularly being given a time and location to attend for confirmatory testing. Mothers in this study valued the screening system of care in mitigating concerns [28].

Conversely, interviews with 87 parents of 44 infants in the US who had been identified as CF carriers following a false positive CF NBS result found that this resulted in poor intra and interpersonal relationships within the family system and more widely. The parents expressed concerns about test accuracy, the child’s health, especially in those who had exhibited signs of respiratory illness, and the future. Parents described the period of uncertainty ending in the child being a carrier rather than being affected by CF, enabling them to gain new perspectives and strengthen their relationship. For one father in the study, the false positive result led to him questioning the child’s paternity. The authors also describe extended family members searching for the source of the genetic defect that had led to the child’s carrier status; wondering if other relatives had CF and/or were carriers. Parents also talked about their support for NBS and feeling empathy for parents of affected children [29]. Interestingly, this study does not mention the time between parents receiving the NBS results and confirmatory testing, which may have mitigated against these negative outcomes [16,27,28].

The relatively recent new designation of CF Screen Positive, Inconclusive Diagnosis (CFSPID) [30] provides another layer of uncertainty for CF NBS. However, there is very little available evidence about the psychological impact of a CFSPID designation on families. A secondary analysis of interview data obtained from a small subset of five couples when their infants were 2 to 6 months old and later at 12 months of age who participated in a larger project demonstrated that uncertainty emerged as the central dimension of parents’ experience when given a CFPSID designation for their child. This uncertainty was linked to the fact that the screening and diagnostic test results were perceived as being contradictory; the presence of two CF mutations from the screening result, usually resulting in an abnormal diagnostic test, CF symptoms, and a CF diagnosis, was confusing to parents, as their child had a normal or borderline sweat test result and were asymptomatic. Moreover, the lack of existing knowledge about the prognostic implications of the identified mutations left health professionals and parents with little certainty about the implications for the infants’ future health trajectory [31]. A more recent study with eight parents (three couples and two mothers) supported these initial findings and suggested that CFSPID results caused parental distress, initiated with the first communication of the result and persisting thereafter, but that approaches to the delivery of CFSPID results might reduce the impact [32]. This supports the findings of studies discussed above regarding the importance of the approach used to deliver positive NBS results for other conditions but, perhaps more importantly, the knowledge of the person delivering an uncertain result and their ability to alleviate the parental anxiety associated with this uncertainty [10,19,21,22].

Whilst being confusing for parents, the unknown longer term implications of certain NBS outcomes can also be challenging for health professionals. For instance, borderline CHT results can be challenging to manage to ensure best outcomes for the child in the longer term [33]. Similarly, a Canadian study identified uncertainty associated with the diagnostic as well as the prognostic outcomes for infants with certain metabolic conditions. Health care providers in this study also described transferring some of the uncertainty to parents while involving them in the ongoing monitoring of their child for signs and symptoms that may indicate a more serious prognostic outcome than initially suspected. Finally, the importance of being honest about uncertainty rather than seeing it as a weakness was also viewed as being important by health care providers [34]. These studies highlight the difficulties faced by parents trying to understand the NBS outcomes of uncertain long-term significance as well as the importance of health professionals have adequate knowledge and skills to manage these conditions and parental expectations.

## 5. Conclusions

The findings of the studies presented above demonstrate the importance of carefully considered information provision to reduce psychological impact when imparting positive CF NBS to parents. The method of delivery of information would seem to be far less important than the knowledge of the person responsible and their ability to answer parents’ questions and provide reassurance [19,21,22,23], particularly if a degree of uncertainty is present, such as with CFSPID results [32]. Despite this, the findings of a recent study found that the CF NBS result is communicated by a range of health professionals internationally and that this may not always be the most appropriate or knowledgeable person but is influenced by many factors, including geographical/logistical, legal, financial and cultural constraints [35]. Additionally, a study in the UK found that specific training for professionals involved in communicating positive NBS results is lacking [36] but is clearly needed to ensure they are adequately prepared to undertake this challenging task. This would also help to meet the suggestions contained within the European guidelines regarding the skills CF team members should possess [12].

Ensuring the most appropriate person communicates a positive CF NBS result is particularly important in cases where there may be a degree of uncertainty, such as for false positive CF NBS results or a CFSPID designation [31,32]. Evidence suggests that good information provision and timely confirmatory testing can mitigate against the long-term psychological distress that has previously been considered to be associated with a false positive CF NBS result [16,27,28].

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
