# Peer review of "Psychological Impact of NBS for CF"

_2409-515X, 2020, doi:10.3390/ijns6020027_

Round 1
Reviewer 1 Report
Thank you for the opportunity to review this draft manuscript. Your effort is appreciated, and your past scholarship is recognized. However, several significant concerns, greater than minor from this reviewers’ perspective, warrant addressing before reconsideration. Commentary will be noted generally and specifically herein.
The topic of psychological impact of NBS is important, including for CF. So too communication by knowledgeable staff. However, little rationale is provided in this manuscript as to whether, or why, the authors consider the impact to differ substantially from any other NBS condition, and if so, explain the factors thought to be involved.
Apart from citing CF specific literature, all based on small sample size, the findings presented may likely hold for most screened conditions, and for most part, potential explanations from the cited authors are missing as well. Even when particular articles compare two conditions and note higher prevalence of some distress, at best that only is a comparison across two conditions. No mention was noted of this limitations, and importantly, no mention was noted of limitations due to small sample size to reach conclusions regarding CF-- a relatively common disorder.
For example, see lines 76+ with hypothyroidism: do the authors believe this finding might holds true for other conditions? Regarding line 90+ with small CF and sickle cell disease groups, there is no mention of whether findings were similar or dissimilar between two conditions
If the authors believe there are inherent distinctions, please describe in introduction and throughout. Similarly, if CF is largely a substitutable example, please state in the intro and then note distinctions later in the literature review whenever you feel CF is most applicable compared with other conditions. In either instance, much clarification is needed: what is the authors’ position and please state why?
Please provide greater balance to the article at the beginning. It is a leap to solely reference the Grob article in the second paragraph and then say “Consequently.” Along with the tradition need to present some diverse references in the introduction, this is particularly so given you only selected a small sample qualitative narrative study that aimed to present a strong oppositional view. It is essential to present alternative views from the onset.
You provide an international overview of the literature, however while not intended, apart from a few transitional words, the body of text is too much of a list without sufficient contrast/compare analysis of the research cited and/or explanation of researcher, or your, thoughts.
Please explain if, and why, responses to Uncertainty may be greater/less than/or similar to other screened conditions. If the impact is the same, this too is important to state.
Please spell out CFSPID in body of text when first appears, along with header title. Not every reader will be familiar, and easy to miss in title.
Literature on 8 parents is noted Line 192 stating means of communicating uncertainty may be helpful to mitigate distress. There is a wealth of data regarding influence of uncertainty in the general NBS literature as well regarding genetic testing in general. Please provide a bit of referenced information. This is an important consideration and by no means exclusive to the CF community.
The Conclusion section is not a conclusion section given all the new literature cited. Consider reorganizing or relabeling.
Author Response
Thank you for the opportunity to review this draft manuscript. Your effort is appreciated, and your past scholarship is recognized. However, several significant concerns, greater than minor from this reviewers’ perspective, warrant addressing before reconsideration. Commentary will be noted generally and specifically herein.
Thank you for your comments, they have been really useful.
The topic of psychological impact of NBS is important, including for CF. So too communication by knowledgeable staff. However, little rationale is provided in this manuscript as to whether, or why, the authors consider the impact to differ substantially from any other NBS condition, and if so, explain the factors thought to be involved.
A paragraph has been added to the introduction addressing this point and lines 51-52.
Apart from citing CF specific literature, all based on small sample size, the findings presented may likely hold for most screened conditions, and for most part, potential explanations from the cited authors are missing as well. Even when particular articles compare two conditions and note higher prevalence of some distress, at best that only is a comparison across two conditions. No mention was noted of this limitations, and importantly, no mention was noted of limitations due to small sample size to reach conclusions regarding CF-- a relatively common disorder.
Additional discussion has been added about the findings of these studies and the implications of these findings e.g. lines 112-116, 122-124, 153-159.
For example, see lines 76+ with hypothyroidism: do the authors believe this finding might holds true for other conditions? Regarding line 90+ with small CF and sickle cell disease groups, there is no mention of whether findings were similar or dissimilar between two conditions
These findings have been elaborated upon in lines 132-137 and lines 139-140.
If the authors believe there are inherent distinctions, please describe in introduction and throughout. Similarly, if CF is largely a substitutable example, please state in the intro and then note distinctions later in the literature review whenever you feel CF is most applicable compared with other conditions. In either instance, much clarification is needed: what is the authors’ position and please state why?
A paragraph addressing this has been added to the introduction and where relevant, this has been addressed throughout the manuscript e.g. lines 51-52.
Please provide greater balance to the article at the beginning. It is a leap to solely reference the Grob article in the second paragraph and then say “Consequently.” Along with the tradition need to present some diverse references in the introduction, this is particularly so given you only selected a small sample qualitative narrative study that aimed to present a strong oppositional view. It is essential to present alternative views from the onset.
As mentioned a paragraph including additional references has been added to the introduction.
You provide an international overview of the literature, however while not intended, apart from a few transitional words, the body of text is too much of a list without sufficient contrast/compare analysis of the research cited and/or explanation of researcher, or your, thoughts.
As mentioned above, additional discussion has been added about the findings of these studies and the implications of these findings e.g. lines 112-116, 122-124, 153-159.
Please explain if, and why, responses to Uncertainty may be greater/less than/or similar to other screened conditions. If the impact is the same, this too is important to state.
An explanation has been added to the beginning of this section, lines 184-189 with additional supporting references.
Please spell out CFSPID in body of text when first appears, along with header title. Not every reader will be familiar, and easy to miss in title.
This has been done.
Literature on 8 parents is noted Line 192 stating means of communicating uncertainty may be helpful to mitigate distress. There is a wealth of data regarding influence of uncertainty in the general NBS literature as well regarding genetic testing in general. Please provide a bit of referenced information. This is an important consideration and by no means exclusive to the CF community.
Additional discussion regarding this has been added in lines 255-266 with additional supporting literature.
The Conclusion section is not a conclusion section given all the new literature cited. Consider reorganizing or relabeling.
Thank you for pointing this out, the paragraph previously included in the conclusion has been moved to lines 88-102 so that it flows better.
Reviewer 2 Report
This is a well-written review on the psychological impact to parents of a positive CF NBS.
My only comment is that it reads as though the health-care professional communicating the NBS result will be the same person who then communicates the diagnostic result (e.g. Abstract lines 14-16), although in many jurisdictions these tasks will be performed by different people. For example, the maternity provider may communicate the NBS result and CF clinical team performs and is responsible for communicating the results of diagnostic tests
There is a typo line 105 "texting' should be "testing."
Author Response
This is a well-written review on the psychological impact to parents of a positive CF NBS.
Thank you for your comment.
My only comment is that it reads as though the health-care professional communicating the NBS result will be the same person who then communicates the diagnostic result (e.g. Abstract lines 14-16), although in many jurisdictions these tasks will be performed by different people. For example, the maternity provider may communicate the NBS result and CF clinical team performs and is responsible for communicating the results of diagnostic tests
Alterations have been made in the abstract lines 14-16) so that it is now clearer that the person giving the NBS result may be different to the person giving the diagnostic result.
There is a typo line 105 "texting' should be "testing."
This has been changed (now line 149). Thank you.
Round 2
Reviewer 1 Report
Thank you for the opportunity to review your manuscript’s revisions.
The issue raised previously in first draft regarding the conclusion section not truly representing a conclusory section is now addressed by moving to section 3. And, a few modifications made also increased clarity, such as adding “The relatively new designation…(CFSPID) provides another layer of uncertainty that is unique to CF NBS.” (line 238-9), and noting differing psychological implications that may arise (guilt) when the NBS finding is of an inherited genetic disorder versus a non-inherited condition.
However, from this reviewer’s perspective, several significant concerns exist, greater than minor, regarding unintentionally misleading information, especially in the Introduction. Some new concerns have emerged with the addition of new material and some that were raised in reviewer comments to first submitted draft have not been adequately addressed.
Some examples of serious concern, including inaccuracies: The new second para Intro (lines 32-40) has several misnomers. In reality, several of the metabolic conditions detected on NBS in some regions can NOT always be managed effectively despite best adherence to diet, meds, or even transplantation – such as MMA null mutation (see Venditti; Vockley; others) on US RUSP. And some, like MMA Mut O, are life limiting with no cure.
The leap into Intro para 2’s last sentence: “Therefore, the psychological impact….differs when compared….” (lines 38-40) is flawed based on example provided (above).
Examples of leaps to support thesis that CF is very distinct from all other conditions remain throughout. For example, and not adequately addressed when commented on in prior review, the next para saying “Qualitative interviews…demonstrated….” (lines 42-48) should clearly state, and upfront, : One small study with ”Qualitative…” Any strong statement attached to strong conclusion (such as “demonstrated”) with small sample needs to have that transparent. And some balance with another article for serious metabolic condition with some similar findings (even lines 151-157) is (and was) recommended to immediately follow or precede current cite 11.
To be balanced, for lines 49-56, recommend also noting a phrase such as Like all conditions, before stating “It is vital…how results communicated” (line 49) and acknowledge that a few others also have the variables “given…..” And note that Like many/some conditions “while it may not be possible….” Then in last line Intro: “this chapter will…” suggest the authors’ note your article will focus on psych impact of CF, with implications not always limited to CF
Given the title is “Psych Impact of CF,” one consideration strongly encouraged is to abandon term “unique,” throughout and just tell the story of what the psych impact is rather than force distinctions as if global in places they do not necessarily exist.Fine to state distinctions , but only when certain they truly exist, and only between specific conditions there is evidence on. Similar when noting commonalities if applicable. I believe these significant issues can ably be addressed by the authors and am happy to review once considered by the authors.
Author Response
The issue raised previously in first draft regarding the conclusion section not truly representing a conclusory section is now addressed by moving to section 3. And, a few modifications made also increased clarity, such as adding “The relatively new designation…(CFSPID) provides another layer of uncertainty that is unique to CF NBS.” (line 238-9), and noting differing psychological implications that may arise (guilt) when the NBS finding is of an inherited genetic disorder versus a non-inherited condition.
Thank you for these helpful suggestions.
However, from this reviewer’s perspective, several significant concerns exist, greater than minor, regarding unintentionally misleading information, especially in the Introduction. Some new concerns have emerged with the addition of new material and some that were raised in reviewer comments to first submitted draft have not been adequately addressed.
Apologies, I think I misunderstood your previous request and thought you wanted me to try and prove CF was unique. I have now amended this accordingly.
Some examples of serious concern, including inaccuracies: The new second para Intro (lines 32-40) has several misnomers. In reality, several of the metabolic conditions detected on NBS in some regions can NOT always be managed effectively despite best adherence to diet, meds, or even transplantation – such as MMA null mutation (see Venditti; Vockley; others) on US RUSP. And some, like MMA Mut O, are life limiting with no cure.
The leap into Intro para 2’s last sentence: “Therefore, the psychological impact….differs when compared….” (lines 38-40) is flawed based on example provided (above).
This has been deleted and amended accordingly.
Examples of leaps to support thesis that CF is very distinct from all other conditions remain throughout. For example, and not adequately addressed when commented on in prior review, the next para saying “Qualitative interviews…demonstrated….” (lines 42-48) should clearly state, and upfront, : One small study with ”Qualitative…” Any strong statement attached to strong conclusion (such as “demonstrated”) with small sample needs to have that transparent.
This has been clarified on lines 39-4, line 41 and line 239.
And some balance with another article for serious metabolic condition with some similar findings (even lines 151-157) is (and was) recommended to immediately follow or precede current cite 11.
This has been amended.
To be balanced, for lines 49-56, recommend also noting a phrase such as Like all conditions, before stating “It is vital…how results communicated” (line 49) and acknowledge that a few others also have the variables “given…..” And note that Like many/some conditions “while it may not be possible….” Then in last line Intro: “this chapter will…” suggest the authors’ note your article will focus on psych impact of CF, with implications not always limited to CF
All of these amendments have been made.
Given the title is “Psych Impact of CF,” one consideration strongly encouraged is to abandon term “unique,” throughout and just tell the story of what the psych impact is rather than force distinctions as if global in places they do not necessarily exist.
The word 'unique' has been removed from the manuscript to provide a more nuanced discussion.
Fine to state distinctions , but only when certain they truly exist, and only between specific conditions there is evidence on. Similar when noting commonalities if applicable. I believe these significant issues can ably be addressed by the authors and am happy to review once considered by the authors.
Thank you
Round 3
Reviewer 1 Report
The modifications undertaken by the authors were beneficial to the accuracy of this manuscript.
A concern remains as to accuracy of what you are implying in your comparison with SCD given the lines 33-36 : {"NBS for CF may pose different challenges when compared to other conditions included in NBS 33 Programmes such as sickle cell disease (SCD) which commonly includes antenatal screening,34 meaning parents are aware of their own carrier status and the theoretical risk to their unborn child 35 [7]. For CF, parents are often unaware of their own carrier status and the disease is considered life 36 limiting; currently, there is no cure [8]} Rephrasing is essential.To be clear, SCD is also considered life limiting, and there are many parents who are unaware they are carriers of SC and many unaware they have SC Trait. Provide a different example if you are unable to clarify this correctly.
Clarification is necessary as well for Lines 46-49; cite 10: "for a metabolic condition" whether your use of singular is only referring to one condition--and if so, state specifically what condition please. Or, whether this should be plural; and then please state "inborn error of metabolism" if applicable (put IEM in parens) or else capitalize Metabolic because "metabolism" was obscured surrounded by other conditions in caps.
Given all the modifications made from version 1, Please modify the abstract accordingly.
And please proofread your document more carefully, or ask a colleague to, as many misplaced comma's remain that are confusing to the reader (ex line 270 "information,") as well as misspellings (ex. line 272 particualrly).
Thank you, and stay well given these challenging times.
Author Response
I have rephrased the sentence about CF and SCD and antenatal testing.
I have changed it to ‘one of the metabolic conditions’ but have not capitalised this.
I don’t think the abstract needs modifying, the message remains the same.
I have proof read it again, corrected the two errors and removed some commas throughout.